# Research and Analysis on Enhancement of Surface Flashover Performance of Epoxy Resin Based on Dielectric Barrier Discharge Plasma Fluorination Modification

**DOI:** 10.3390/nano14171382

**Published:** 2024-08-24

**Authors:** Xizhe Chang, Yueyi Sui, Changyu Li, Zhanyuan Yan

**Affiliations:** 1Department of Mathematics and Physics, North China Electric Power University, Baoding 071000, China; 18231319118@163.com; 2State Grid Hebei Electric Power Co., Ltd. Ultra High Voltage Branch, Shijiazhuang 050070, China; suiyueyi2024@163.com; 3College of Electrical and Electronic Engineering, North China Electric Power University, Beijing 102206, China; l13925894863@163.com; 4Hebei Key Laboratory of Physics and Energy Technology, North China Electric Power University, Baoding 071000, China

**Keywords:** epoxy resin, plasma, SF_6_, surface flashover voltage, charge traps

## Abstract

To conquer the challenges of charge accumulation and surface flashover in epoxy resin under direct current (DC) electric fields, numerous efforts have been made to research dielectric barrier discharge (DBD) plasma treatments using CF_4_/Ar as the medium gas, which has proven effective in improving surface flashover voltage. However, despite being an efficient plasma etching medium, SF_6_/Ar has remained largely unexplored. In this work, we constructed a DBD plasma device with an SF_6_/Ar gas medium and explored the influence of processing times and gas flow rates on the morphology and surface flashover voltage of epoxy resin. The surface morphology observed by SEM indicates that the degree of plasma etching intensifies with processing time and gas flow rate, and the quantitative characterization of AFM indicates a maximum roughness of 144 nm after 3 min of treatment. Flashover test results show that at 2 min of processing time, the surface flashover voltage reached a maximum of 19.02 kV/mm, which is 25.49% higher than that of the untreated sample and previously reported works. In addition to the effect of surface roughness, charge trap distribution shows that fluorinated groups help to deepen the trap energy levels and density. The optimal modification was achieved at a gas flow rate of 3.5 slm coupled with 2 min of processing time. Furthermore, density functional theory (DFT) calculations reveal that fluorination introduces additional electron traps (0.29 eV) and hole traps (0.38 eV), enhancing the capture of charge carriers and suppressing surface flashover.

## 1. Introduction

Epoxy resin plays a crucial role in applications such as power electronics, aerospace technology, and automotive industries owing to its excellent insulation performance, outstanding mechanical strength, and favorable chemical corrosion resistance [1,2,3,4,5,6]. Nevertheless, epoxy resin as insulation is prone to charge accumulation under high direct current (DC) fields, causing partial discharge and surface flashover [7,8,9,10], seriously threatening the service life of the devices [11,12]. Hence, improving the flashover performance of epoxy resin is significant for the stable operation of electrical equipment [13,14].

To address these issues, adding inorganic nanoparticles is considered an effective strategy [15,16]. The wide bandgap of inorganic fillers facilitates the formation of charge traps within the epoxy resin matrix, suppressing carrier migration and hindering flashover development [17]. Xue et al. coated fluorinated nano SiC on the surface of epoxy resin and found that it effectively accelerated the charge dissipation [18]. Lv et al. studied different contents in Al_2_O_3_ fillers on the breakdown voltage of the epoxy resin, and the mechanism of performance improvement was analyzed through traps theory [19]. Mai et al. introduced BaTiO_3_ particles into epoxy resin and found that the optimal breakdown strength was achieved with 10 wt% doping concentrations [20]. However, such a doping strategy has incurred some problems. On the one hand, nanofillers are prone to agglomeration, which can cause damage to the polymer matrix [21,22]. On the other hand, the doping of fillers inside the matrix cannot ensure uniform dispersion of charge traps on the surface, and the surface flashover problem remains unsolved. Plasma treatment has the advantages of short-term high efficiency, energy conservation, and environmental protection [23,24,25]. This method can change the surface physicochemical properties of materials and effectively improve their surface flashover characteristics [26,27]. Dielectric barrier discharge (DBD) is one of the plasma treatment methods [28] which has the advantages of uniform discharge and minimal damage to the material itself [29]. In addition, fluorine (F), with high electronegativity, positively impacts the electrical performance of insulation materials by affecting discharge development [30]. Therefore, fluorinated gas can be used as the medium for dielectric barrier discharge plasma treatment of insulation materials to enhance their surface flashover. Zhang et al. used the CF_4_ plasma method to treat meta-aramid insulating paper, and the results showed that appropriate plasma treatment can increase the surface roughness of the material, increasing the creepage distance of surface flashover, and the formation of a fluoride layer can further suppress charge accumulation [31]. Ma et al. performed the DBD method to treat LDPE samples by applying a microsecond pulse power supply. The hydrophilicity of the sample was effectively enhanced, and the problem of charge accumulation was solved [32]. Shao et al. used the DBD method to fluorinate PMMA, which increased the surface roughness and successfully introduced physical traps. Additionally, the grafted fluorinated groups further impeded secondary electron emission and enhanced the flashover voltage [33].

The above research indicates that appropriately increasing surface roughness and introducing fluorine groups can effectively improve the surface flashover voltage. Nevertheless, current DBD plasma treatment primarily utilizes CF_4_/Ar as the medium gas [34,35]. Although SF_6_/Ar as an efficient medium gas has been widely used in inductively coupled plasma treatment due to its superior etching rate and abundant fluorine groups [36,37,38,39], its application in DBD has received scant attention and remains largely unexplored. In addition, current research suggests that surface roughness from plasma etching is the main factor improving surface flashover, while the contribution and mechanism of charge traps generated by fluorinated groups remain unclear.

In this work, the DBD plasma discharge device with SF_6_/Ar is designed, and the mechanism of surface fluorination modification of epoxy resin is studied. The effects of different processing times and gas flow rates on the surface morphology and roughness are observed. The surface flashover voltage is tested and the correlation between roughness and flashover is analyzed. Moreover, the charge trap distribution characteristics on the epoxy resin surface under different fluorination conditions are tested using the isothermal surface potential decay (ISPD) method. Finally, the underlying mechanism for the improvement in surface flashover is explained by density functional theory (DFT) calculation, providing theoretical guidance for plasma fluorination-modified insulation materials to enhance surface flashover.

## 2. Materials and Methods

### 2.1. Materials

The epoxy resin samples used in this work are provided by Shandong Taikai High Voltage Switchgear Co., Ltd., (Taian, China). Among them, Diglycidyl ether of Bisphenol A (DGEBA) and methyl tetrahydro phthalic anhydride (MTHPA) are utilized as the epoxy resin matrix and curing agent, respectively, while Al_2_O_3_ particles with a size of approximately 500 nm are incorporated as the reinforcing fillers. The mass ratio of the three components (DGEBA:MTHPA:Al_2_O_3_) is 100:38:330. The sample size is a circle with a diameter of 50 mm and a thickness of 5 mm. Before the experiment, the epoxy resin sample is wiped and then cleaned with anhydrous ethanol and deionized water in an ultrasonic cleaning machine to remove the surface dirt. Finally, the epoxy resin sample is placed in a vacuum-drying oven and dried for 48 h. After the sample is dried, it is taken out for subsequent experiments. The used gases Ar and SF_6_ are purchased from Baoding North Special Gas Company (Baoding, China), with a purity of 99.99%.

### 2.2. DBD Plasma Experimental Devices

Figure 1 illustrates the schematic diagram of the plasma experimental setup, which consists of a pneumatic system and a circuit system. The pneumatic system primarily comprises SF_6_ and Ar gases, a digital flowmeter, and a collection device, where SF_6_ serves as the reaction gas to supply fluorine and Ar is chosen as the carrier gas. The gas flow rate is controlled by the D07-7C digital flowmeter provided by Beijing Sevenstar Huachuang Electronics Co., Ltd. (Beijing, China). The gas circuit is used to connect the gas source, digital flowmeter, and reactor, allowing the gas to flow into the reactor. Notably, the outlet of the reactor is equipped with a seamlessly integrated collection device, ensuring effective recovery of the introduced SF_6_ gas. The recovered SF_6_ is transferred to a professional hazardous waste agency to ensure that it is not directly released into the air. Additionally, the circuit system consists of a high-frequency power supply, an oscilloscope, and a plasma reactor. The high-frequency power supply (CTP-2000K, Nanjing Suman Plasma Technology Co., Ltd., Nanjing, China) is utilized for generating plasma. It boasts a central frequency of 50 kHz, an output voltage range of 0~30 kV, and a maximum output power of 500 W. Equipped with a 1000:1 voltage divider, the actual output voltage can be measured through the front panel interface, which displays the voltage division ratio. The plasma reactor (DBD-50, Nanjing Suman Plasma Technology Co., Ltd., Nanjing, China) features upper and lower circular metal electrodes with a diameter of 50 mm. The upper electrode is connected to the power supply through a high-voltage connection terminal, and the lower electrode is connected to the ground. The discharge surface has a diameter of 50 mm, and the discharge gap can be adjusted from 0~10 mm. The plasma quartz dielectric reactor (DBD-100B, Nanjing Suman Plasma Technology Co., Ltd., Nanjing, China) has a central reaction area with a depth of 3 mm and a diameter of 50 mm. The upper dielectric plate is made of quartz glass with a thickness of 2 mm. The oscilloscope (DPO 2024B, Nanjing Suman Plasma Technology Co., Ltd., Nanjing, China) is used to measure the current and voltage output of the power supply during the reaction. The entire experimental setup is provided by Nanjing Suman Plasma Technology Co., Ltd., (Nanjing, China).

### 2.3. Fluorination Treatment Process with Plasma

Firstly, the reaction vessel is wiped using anhydrous ethanol, and the epoxy sample is placed into the reactor. The volume of the sample is controlled to within half that of the reaction vessel to ensure the normal flow of SF_6_ and Ar gas. Secondly, the reaction vessel is placed in the middle of the device and tightly connected to the electrodes by adjusting the electrode spacing. The SF_6_ gas flow rate is adjusted to 0.5 slm (standard liter per minute). As the SF_6_ gas sufficiently flows into the reaction vessel, Ar gas is introduced with a flow rate of 5 slm. Thirdly, the output voltage from the high-frequency AC power supply is set as 4.5 kV, and the frequency is 50 kHz. When a purple glow discharge appears in the reaction vessel, we start recording the time. According to previous experimental experience, the Ar ventilation rate is set as 5 slm, and that of the SF_6_ is set as 3.5 slm. The plasma processing times are set as 1, 2, 3, 4, and 5 min, respectively. The obtained samples are marked as T1, T2, T3, T4, and T5. Additionally, the untreated epoxy resin sample, annotated as T0, serves as the control.

### 2.4. Characterizations

*X*-ray photoelectron spectroscopy (XPS, ThermoFisher Nexsa, Waltham, MA, USA) is utilized to observe the fluorination effect on epoxy resin. A scanning electron microscope (SEM, BRUKER XFlash 6130, Bremen, Germany) and atomic force microscope (AFM, Bruker BRUKER Dimension Icon, Bremen, Germany) are employed to observe the surface morphology variation after plasma treatment.

### 2.5. Performance Testing

A pair of interfinger electrodes are utilized to test the surface flashover voltage. The electrodes are made of brass with a head arc radius of 7.5 mm. The sample is placed on a PTFE insulated bracket and the electrodes are connected to a DC power supply. During the test, a positive DC voltage is applied to the sample at a boost rate of 1 kV/s until surface flashover occurs. The flashover voltage *U*_B_ is tested by 10 sets of data per sample and analyzed using a two-parameter Weibull model, by which the *U*_B_ is extracted from the breakdown possibility of 63.2%.

The isothermal surface potential decay (ISPD) method is implemented to observe the charge trap characteristics, for which the platform is displayed in the literature [40]. During the test, the sample is charged by a corona needle for 120 min. Subsequently, an electrostatic probe (Trek P0865 connected to Trek 3455ET, Denver, CO, USA) is used to collect the surface potential of the sample for 720 s. The surface potential (*U*_s_) as a function of time (*t*) is recorded by the acquisition card (Trek, Denver, CO, USA). The trap density (*E*_T_) and trap intensity (*I*_T_) are obtained using Equations (1) and (2) [41],
*E*_T_ = *k*_B_*T*ln(*vt*)(1)
(2)$$I_{T}=\frac{\varepsilon_{0}\varepsilon_{r}}{et_{f}}\times t\frac{dU_{S}}{dt}$$
where *k*_B_ is the Boltzmann constant, *T* represents the temperature, *v* is the carrier escape frequency with a value of 4.17 × 10^13^ s^−1^, *e* is the unit electron charge being 1.60 × 10^−19^ C, *ε*_0_ is the vacuum permittivity being 8.85 × 10^−12^ F/m, and *ε*_r_ is the relative dielectric constant.

## 3. Results

As shown in Figure 2, the surface of the untreated epoxy resin (T0) exhibits C 1s and O 1s peaks. After DBD plasma treatment, an additional F 1s peak is detected on the surface [42], indicating that the DBD treatment altered the chemical properties of the epoxy resin and facilitated the grafting of fluorinated functional groups onto its surface. Furthermore, as the processing time increases, the peak intensity of F 1s gradually rises, suggesting an enhancement in the fluoride grafting rate.

As shown in Figure 3a, the surface of the untreated epoxy resin sample exhibits a relatively smooth and compact morphology. Compared with the plasma-treated samples (T1~T5), the surface morphology of T0 has dense features, and no particle accumulation is observed on the surface. As the processing time increases, the plasma continuously etches the substrate of the sample, and the Al_2_O_3_ fillers inside the matrix are gradually exposed. In Figure 3b, there is almost no obvious influence on the surface as the processing time is less than 1 min. In Figure 3c, when the processing time reaches 2 min, the surface cracks are observed and the inside Al_2_O_3_ particles begin to emerge and accumulate. As the processing time continues to increase in Figure 3d–f, more and more fillers leak out. The number and distribution area of Al_2_O_3_ particles present a significant increase over time, causing severe damage to the epoxy resin matrix.

The element mapping is performed using EDS and the mapping image of T5 is shown in Figure 3g–i. It is evident that the Al element in Figure 3g is attributed to the Al_2_O_3_ particles, and the F element and S element in Figure 3h,i illustrate the effective treatment of SF_6_ on the epoxy surface.

The roughness of the sample surface after plasma etching is observed by AFM in Figure 4. The surface of the untreated epoxy resin presents a relatively smooth state, and the roughness *R*_q_ is only 61.4 nm. At the initial stage of plasma treatment, the *R*_q_ of epoxy resin begins to increase with processing time, exhibiting obvious grooves and protrusions on the surface of the sample. When the processing time is 1 min, the *R*_q_ is increased to 95.5 nm, owing to the exposure of a small amount of Al_2_O_3_ particles. As the processing time reaches 3 min, the *R*_q_ has a maximum value of 144 nm, suggesting the highest surface roughness. Moreover, as the processing time continues to increase, the *R*_q_ tends to decrease. This may be because the exposed Al_2_O_3_ particles after plasma etching accumulate on the surface, contributing to filling the depressed area to some extent [43,44].

Figure 5a shows the Weibull distribution of the breakdown probability of flashover voltage for each sample, and the flashover voltage is extracted from the value at a breakdown probability of 63.2%, which is depicted in Figure 5b. The plasma treatment with SF_6_ contributes to enhancing the surface flashover voltage of epoxy. When the processing time is 2 min, the flashover voltage increases to 19.02 kV/mm, which is 25.49% higher than that of the pristine sample. Notably, this improvement in surface flashover voltage has demonstrated superiority over traditional DBD plasma treatment using a CF_4_ gas medium [26,45,46]. This can be attributed to two aspects: one is that the appropriate increase in surface roughness is not conducive to the local accumulation of charges, accelerating the dissipation of surface charges. As shown in Figure 4, T2 possesses a relatively high value of *R*_q_, which makes it more difficult for charges to accumulate on the surface for a long period, effectively suppressing the local electric field distortion and enhancing the surface flashover voltage. Additionally, the other important factor is the further introduction of deep charge traps on the surface of the epoxy resin by plasma treatment. Figure 5c demonstrates the distribution of surface charge traps of various samples. The pure epoxy resin has two trap peaks, corresponding to the shallow and deep traps, respectively. After plasma treatment, the shallow trap peak of the sample significantly weakened, while the energy level and density of the deep trap peak continued to increase. Considering that the deep traps play a major role in capturing charges and impeding the formation of leakage current, the maximum peak values of deep traps in various samples are extracted and compared in Figure 5d. Corresponding to the flashover voltage, the T2 sample also exhibits the highest deep trap energy level and density. Due to the strong electronegativity of fluorine, the grafting of derivative groups of SF_6_ onto the surface of the epoxy resin is beneficial for the generation of deep traps. However, as the processing time increases, the exposed Al_2_O_3_ particles aggregate on the surface, which affects the uniform distribution of fluorinated groups on the epoxy resin. In addition, the aggregates of fillers severely degrade the surface structure, resulting in the reappearance of shallow traps, which is not conducive to the improvement in flashover voltage.

During the plasma treatment, in addition to the processing time, the gas flow rate also contributes as an important factor, determining the grafting rate of functional groups. Hence, to further explore the optimal plasma treatment conditions under SF_6_ gas, the influence of different gas flow rates under 2 min is further investigated. The choice of 2 min is based on the consideration that such a configuration has the greatest increase in surface flashover voltage. Moreover, considering that the SF_6_ gas flow rate was set as 3.5 slm earlier, different rate gradients of 1.5, 2.5, 3.5, and 4.5 slm are selected for comparative analysis. The samples treated with different gas flow rates are annotated as F1.5, F2.5, F3.5, and F4.5.

As shown in Figure 6, the surface roughness of the epoxy resin continuously increases with SF_6_ gas flow rates. When the gas flow rate reaches 4.5 slm, the exposed Al_2_O_3_ particles undergo significant agglomeration, forming larger agglomerations on the surface of the epoxy resin, which has a deterioration effect on the matrix structure.

The surface roughness is further quantitatively characterized using AFM in Figure 7, and the results show that with the increase in the SF_6_ gas flow rate, the surface roughness continues to increase and reaches the maximum value of 128 nm under 4.5 slm.

It can be seen in Figure 8a that the gas flow rates of SF_6_ have varying effects on the surface flashover voltage of the samples. The flashover voltage of the sample in Figure 8b initially increases and then decreases with the gas flow rate, and the maximum is reached at 3.5 slm. The distribution of charge traps in Figure 8c also indicates that upon this configuration, the epoxy resin sample develops the deepest trap energy level and density, being conducive to hindering the charge transport and further improving the flashover voltage. Notably, as the flow rate of SF_6_ gas continues to increase to 4.5 slm, both the flashover voltage and trap depth exhibit degradation. This is due to excessive fluorination causing damage to the molecular structure of the epoxy resins. To further reveal the underlying mechanism, several types of molecular conformation and energy band structures with different degrees of fluorination are studied using the density functional theory (DFT) method.

As shown in Figure 9a, epoxy resin (EP) is generated by the cross-linking reaction of DGEBA and MTHBA monomers. Subsequently, the appropriate and excessive fluorination processes are simulated, respectively, as shown in Figure 9b,c. During appropriate fluorination, due to the strong electronegativity of fluorine, the F atoms on SF_6_ seize the H atoms on EP to form HF (as indicated by the blue arrow), while the remaining F atoms form C–F bonds with unsaturated C (as indicated by the red arrow). However, during excessive fluorination, in addition to the formation of C–F bonds, the F atom continues to capture the H atom on the hydroxyl group, resulting in the grafting of the SF group as well [47].

Geometric optimization is performed using Gaussian 16.0 on the three molecular structures to achieve a reasonable conformation. After that, a single-point energy calculation is conducted to further refine the structural energy for subsequent property analysis. During the calculation, the hybrid function and group basis set are selected as B3LYP and the 6-31G(d), respectively. The calculated results are imported into Gauss View 6.0, and the visualization of HOMO (Highest Occupied Molecular Orbital) and LUMO (Lowest Unoccupied Molecular Orbital) is achieved from the molecular orbital (MO) module [48]. The obtained results are depicted in Figure 10.

According to the frontier molecular orbital theory [49], HOMO (Highest Occupied Molecular Orbital) and LUMO (Lowest Unoccupied Molecular Orbital) are commonly used to represent the energy levels at which a molecule loses and gains electrons. In Figure 10a, the EP has a HUMO of −5.64 eV and a LUMO of −0.56 eV. After appropriate fluorination, –F groups are grafted onto the EP molecular chains, causing a decrease in both the LUMO and HOMO energy levels in Figure 10b. As shown in Figure 10c, excessive fluorination is introduced into the –SF groups, which significantly affects the band structure, resulting in a significant decrease in LUMO and an increase in HOMO.

For an intuitive comparison, the schematic diagrams of the band structures of the three substances are demonstrated in Figure 11a. Compared with EP, F–EP presents an increased bandgap, which is beneficial for improving the insulation ability of the composites. Nevertheless, the bandgap of S–EP is only 0.36 eV, and such a narrow bandwidth would not be able to effectively capture transport carriers, causing significant degradation to its insulating properties. This also explains the flashover voltage deterioration phenomenon caused by the excessive flow rate of the SF_6_ gas mentioned above. Moreover, the alignment arrangement of the band structure is displayed in Figure 11b. The introduction of fluorinated groups endows EP and F–EP with different band arrangement positions, and hence, the band alignment equilibrium would result in the mismatch of HOMO and LUMO levels, forming interfacial electronic traps (*Φ*_e_ = 0.29 eV) and hole traps (*Φ*_h_ = 0.38 eV) [50]. The establishment of electronic and hole traps is conducive to improving the capture of carriers, further hindering the development of leakage current, which also reveals the underlying mechanism for the enhancement in flashover voltage with fluorination.

## 4. Conclusions

This article proposes a surface fluorination treatment strategy for epoxy resin using the DBD plasma method with SF_6_/Ar medium gas and explores the improvement effect of different fluorination durations and gas flow rates on the surface flashover voltage. The obtained conclusions are as follows:

The results indicate that using SF_6_/Ar as the medium gas in DBD plasma treatment exhibits significant etching effects and demonstrates the effective grafting of fluorinated groups. SEM observation shows that surface etching intensifies with processing time and gas flow rate. Through the quantitative characterization of an AFM, roughness is observed to initially increase and then decrease with processing time, reaching a maximum roughness of 144 nm after 3 min of treatment. The flashover test results indicate that at 2 min of processing time, the surface flashover voltage reaches a maximum of 19.02 kV/mm, which is 25.49% higher than that of the pristine sample. This also means that roughness is not the only factor determining flashover voltage. The charge trap distribution characteristics indicate that the introduction of fluorinated groups deepens the charge trap energy levels and effectively improves the trap density, being consistent with the variation in flashover voltage. In addition, the influence of different gas flow rates on the morphology and flashover characteristics of materials is investigated. The results show that the surface roughness of epoxy resin continuously increases with the SF_6_ gas flow rate, and the optimal flashover voltage is realized under a configuration of 3.5 slm and 2 min treatment. The underlying mechanism for the improvement in surface flashover resistance is revealed through DFT calculation and energy band structure analysis. Fluorine group grafting alters the band structure of epoxy resin, introducing electron traps (0.29 eV) and hole traps (0.38 eV), which are conducive to hindering charge carriers and suppressing surface flashover. This provides theoretical guidance for designing and modifying high-performance epoxy resin insulation materials.

## Figures and Tables

**Figure 1 nanomaterials-14-01382-f001:**
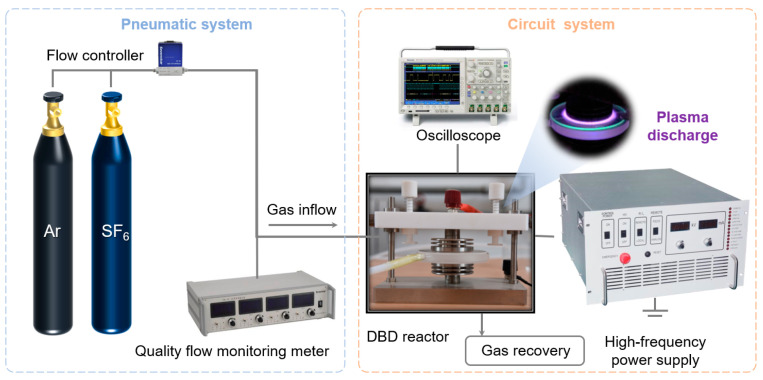
Schematic diagram illustrating DBD plasma experimental setup with SF_6_/Ar gas.

**Figure 2 nanomaterials-14-01382-f002:**
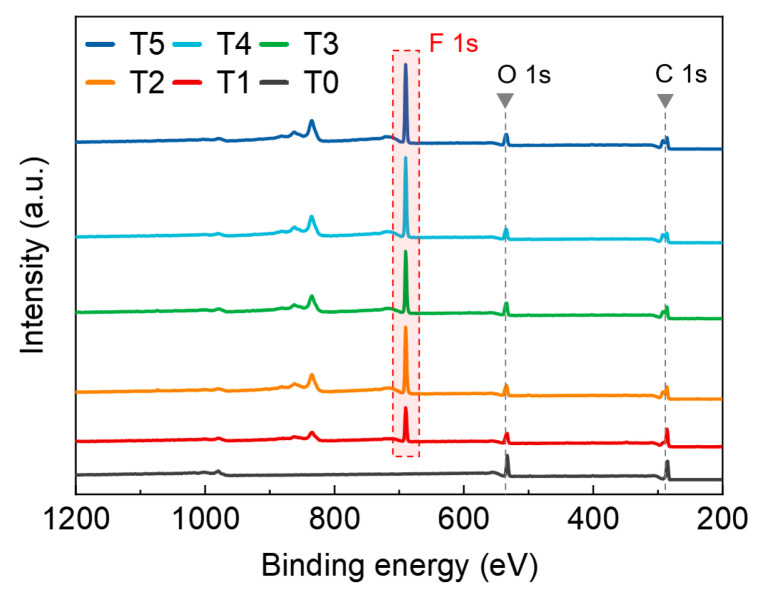
The XPS spectra of samples with various processing times.

**Figure 3 nanomaterials-14-01382-f003:**
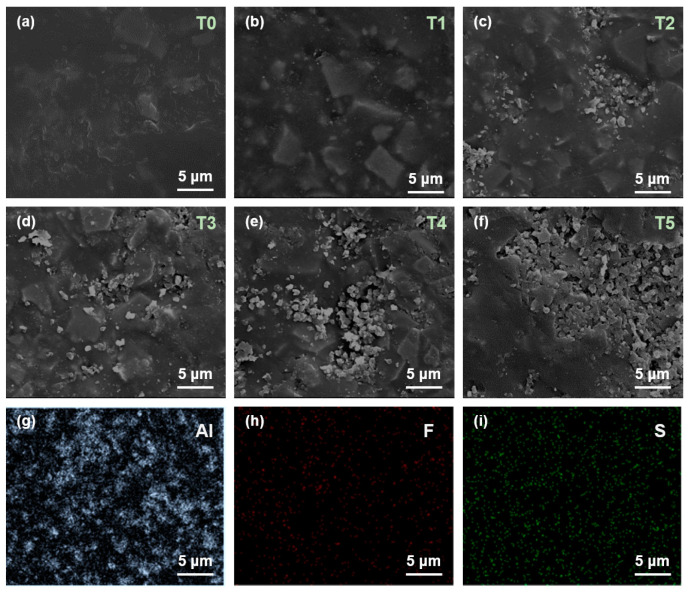
SEM image of surface morphology of (**a**) T0, (**b**) T1, (**c**) T2, (**d**) T3, (**e**) T4, and (**f**) T5. Element mappings extracted from T5 of (**g**) Al, (**h**) F, and (**i**) S.

**Figure 4 nanomaterials-14-01382-f004:**
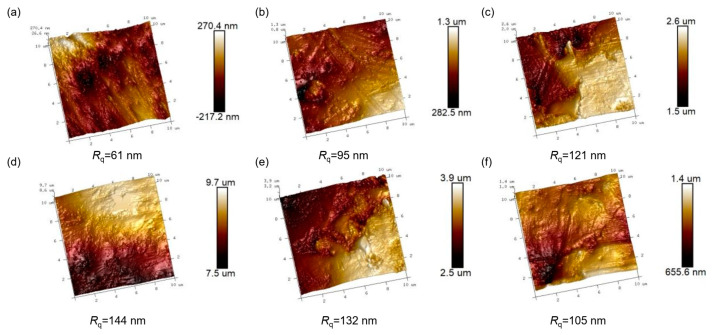
Surface AFM images of (**a**) T0, (**b**) T1, (**c**) T2, (**d**) T3, (**e**) T4, and (**f**) T5.

**Figure 5 nanomaterials-14-01382-f005:**
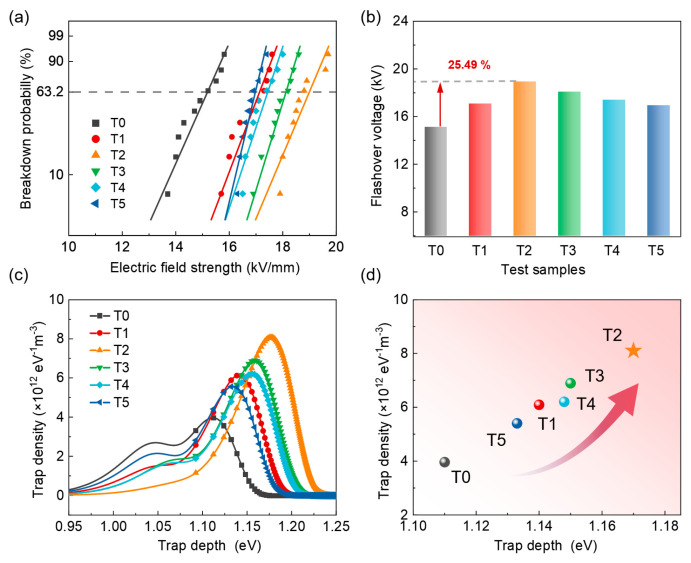
(**a**) The Weibull distribution of the breakdown probability of flashover voltage. (**b**) The flashover voltage of various samples extracted from (**a**). (**c**) The charge trap distributions of various samples. (**d**) The comparison of deep trap peaks of various samples.

**Figure 6 nanomaterials-14-01382-f006:**
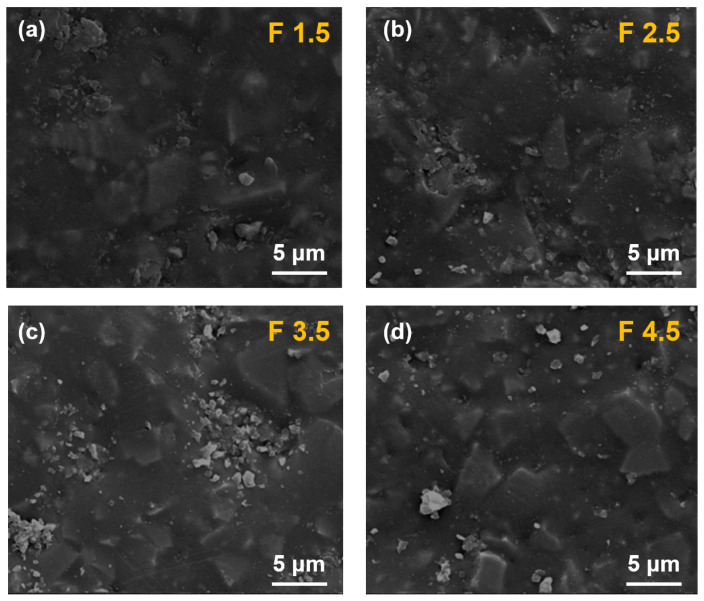
SEM image of surface morphology of (**a**) F1.5, (**b**) F2.5, (**c**) F3.5, and (**d**) F4.5.

**Figure 7 nanomaterials-14-01382-f007:**
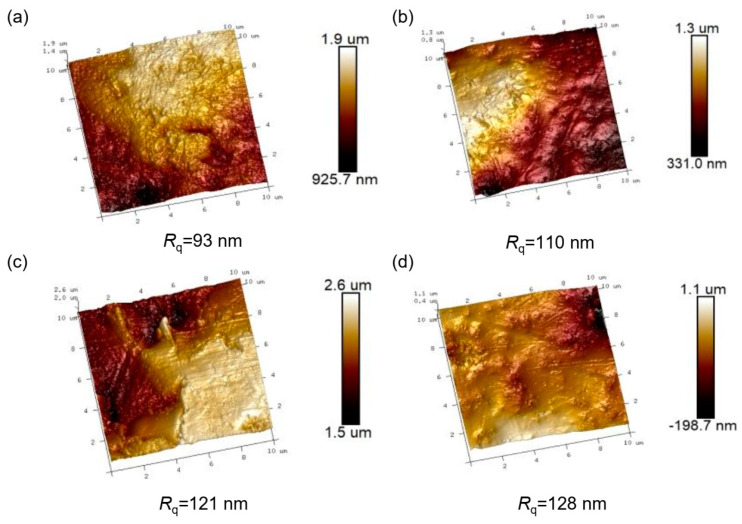
AFM image of surface morphology of (**a**) F1.5, (**b**) F2.5, (**c**) F3.5, and (**d**) F4.5.

**Figure 8 nanomaterials-14-01382-f008:**
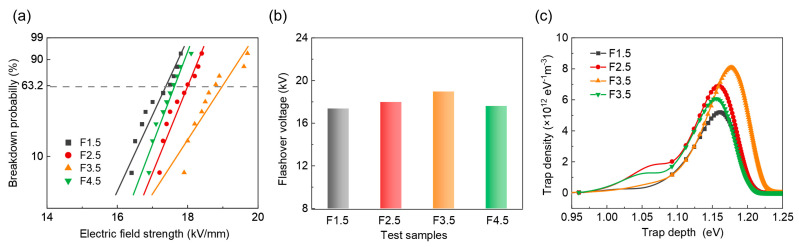
(**a**) The Weibull distribution of the breakdown probability of flashover voltage. (**b**) The flashover voltage of various samples extracted from (**a**). (**c**) The charge trap distributions of the samples.

**Figure 9 nanomaterials-14-01382-f009:**
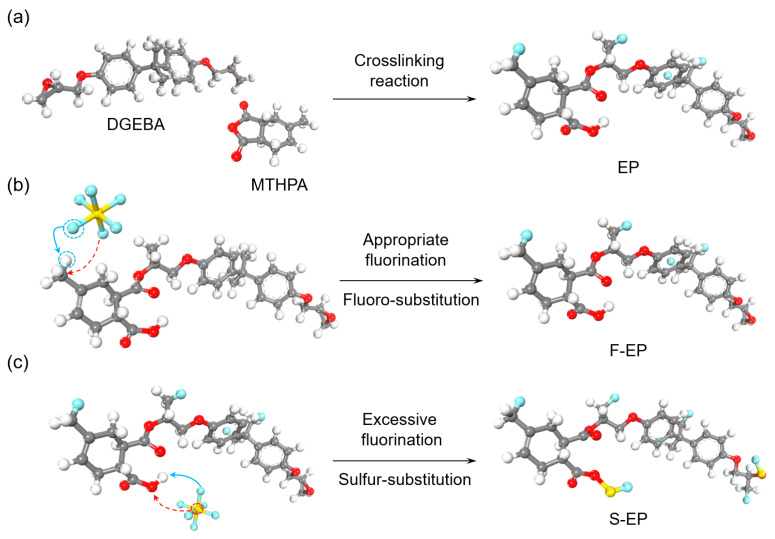
The chemical reaction and formation of (**a**) EP, (**b**) F–EP, and (**c**) S–EP.

**Figure 10 nanomaterials-14-01382-f010:**
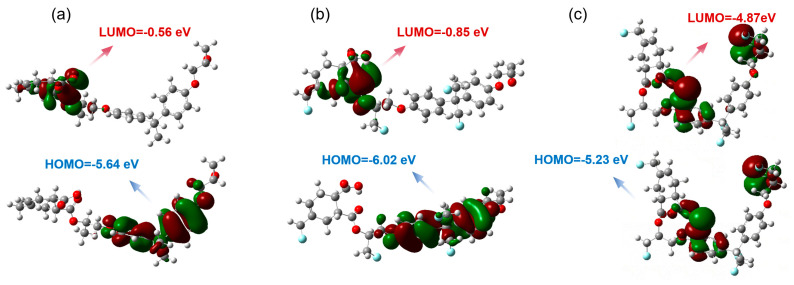
The HUMO and LUMO energy levels of (**a**) EP, (**b**) F–EP, and (**c**) S–EP.

**Figure 11 nanomaterials-14-01382-f011:**
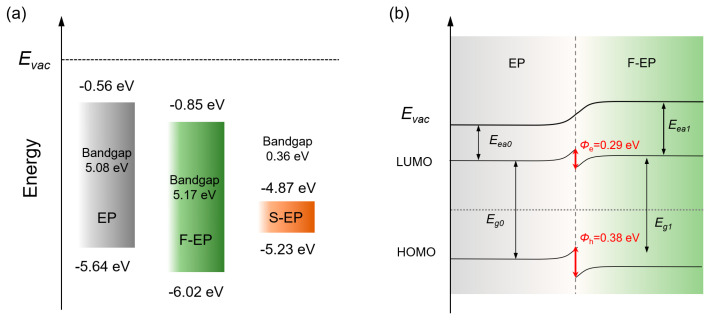
(**a**) A schematical diagram illustrating the energy band structure of EP, F–EP, and S–EP. (**b**) The energy band alignment of EP and F–EP.

## Data Availability

The data will be made available on request.

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
