# Peer review of "Research and Analysis on Enhancement of Surface Flashover Performance of Epoxy Resin Based on Dielectric Barrier Discharge Plasma Fluorination Modification"

_nanomaterials, 2024, doi:10.3390/nano14171382_

Round 1

Reviewer 1 Report

Comments and Suggestions for Authors

The present manuscript by Yan et al. deals with the surface functionalization of epoxy resins by plasma-assisted methods. The aim is the improvement of the surface flashover performance in electrical enginnering applications.

Although this topic is of interest for readsers in this field, the manuscript suffers from serious deficiencies:

# Section 2.1: no imformation is provided about the type and chemistry of the epoxy material, and the applied curing method. Moreover, no statement is given on the type and content of filler, and the particle size of the filler.

This is also weird because the authors apply molecular modelling (DFT) in a later section of their manuscript.

# Section 2.4: The authors write that XPS spectroscopy was applied to analyze the atomic composition of the plasma-treated epoxy resins. However, no such results are presented in the manuscript.

# Section 2.5: reference is made to two equations (3) and (4) but no equation at all appears in the manuscript.

# Apart from that, other deficiencies are to be ststed:

The introduction is quite long but contains repeating sentences and redundant information

Some explanations are rather speculation, and are not backed by results, e.g., lines 186 - 188. Also the assumption in lines 253 - 254 (plasma damage to the surface) is not backed by results, and chemical information is missing.

Finally it remains unclear hoew this topic is related to nanotechnology and nanomaterials, as the filler size is obviously not in the nm range.

For all these reasons I do not recommend this manuscript for publication.

Comments on the Quality of English Language

The quality of English Language is OK, but needs refinement and improvement.

Author Response

We appreciate your professional comments on this article and have prepared a point-to-point response. Please refer to the attached file for further details.

Reviewer 2 Report

Comments and Suggestions for Authors

The paper is devoted for research and analysis on the enhancement of surface flashover performance of epoxy resin based on DBD plasma fluorination modification. However, the paper contains unexplained places (below) and need major revisions.

Abstract should be rewritten in more logic way. For example, please check the sentence at lines 20-23.

At line 153 it was mentioned equations 3-4, however I not find any equations in the paper.

DBD plasma experimental setup (Fig. 1) should be explained more in details.

Lines 160-161, please explain term ‘’a relative smooth and compact morphology’’.

Lines 256-257 and 270-271 corresponding references should be added.

Figs. 8-10 more details about calculations technique should be added.

English need minor revisions.

Comments on the Quality of English Language

English need minor revisions, the main problem article the use.

Author Response

(The authors gave the same response as above.)

Reviewer 3 Report

Comments and Suggestions for Authors

The manuscript explores the enhancement of surface flashover performance in epoxy resin through dielectric barrier discharge (DBD) plasma fluorination modification. By varying treatment times and gas flow rates of SF6/Ar, the study analyzes the impact on surface morphology and flashover voltage. The results show that optimal fluorination conditions significantly improve flashover voltage, attributed to the introduction of charge traps and surface roughness.  However, the use of SF6 in this study raises environmental concerns due to its high global warming potential. To be recommend for publishing in Nanomaterials MDPI, the authors should address how they plan to handle and dispose of SF6 safely, including measures to prevent leaks and strategies for recycling or neutralizing the gas, to minimize its environmental impact.

Comments on the Quality of English Language

 The manuscript requires some moderate editing of English language to enhance clarity and readability including the structure of sentences, Grammer and choice of words. Here are some examples but not limited to improve:

1.     "The surface roughness showed a variation trend of first increasing and then decreasing via processing time..."

Suggeste to : "Surface roughness initially increased and then decreased with processing time..."

2.     "The results shows that the surface roughness of epoxy resin continuously increases via the SF6 gas flow rate..."

 Suggested: "The results show that the surface roughness of epoxy resin continuously increases with the SF6 gas flow rate..."

3.     "The flashover test results show that as the processing time is 2 min, the surface flashover voltage of increases to the maximum of 19.02 kV/mm..."

Suggest to: "The flashover test results indicate that at 2 minutes of processing time, the surface flashover voltage reaches a maximum of 19.02 kV/mm..."

Author Response

(The authors gave the same response as above.)

Reviewer 4 Report

Comments and Suggestions for Authors

Review

Some suggestions to enhance the reader's interest:

·       •It is crucial to include more literature on the topic of "epoxy resin" due to its significance.

·       This reason is rarely discussed. "SF6/Ar" is of significant relevance. It is necessary to include literature and engage in some discussion in the abstract and conclusions.

·       The frequent mention of "flashover voltage" necessitates including references to establish the international positioning of this paper.

·       This physical property, "charged traps," requires discussion in the abstract and conclusion and the addition of relevant literature.

·       This physical property, "surface flashover voltage" needs to be included in the abstract, along with some relevant references.

·       The abstract and conclusions do not discuss the important results obtained from the SEM image.

·       The abstract does not discuss the results of the "AFM image," and it needs to include conclusions.

·       The abstract and conclusions must discuss the Weibull distribution and include a literature review.

Comments on the Quality of English Language

No comments

Author Response

(The authors gave the same response as above.)

Round 2

Reviewer 1 Report

Comments and Suggestions for Authors

The authors have revised their manuscript thoroughla, and have followed the suggestions of the reviewers. Now that XPS spectra, sufficient chemical information on the epoxy resins, and the missing formulas have been added, the manuscript is in mauc better shape than the original version.

Some minor improvemnts in English language still remain to be done. E.g., do not write "reaction kettle", but "reaction vessel" or "rection tube".

The paper will be ready for publication after these improvements.

Comments on the Quality of English Language

English is OK, but minor editing is still necessary.

Author Response

Comments 1:

The authors have revised their manuscript thoroughly, and have followed the suggestions of the reviewers. Now that XPS spectra, sufficient chemical information on the epoxy resins, and the missing formulas have been added, the manuscript is in much better shape than the original version.

Some minor improvements in English language still remain to be done. E.g., do not write “reaction kettle”, but “reaction vessel” or “rection tube".

The paper will be ready for publication after these improvements.

Response 1:

We appreciate the positive comments made by the reviewer and sincerely value your suggestions, which have improved our manuscript and helped it meet the standards for publication. According to your latest suggestion, we have revised the expression of “reaction vessel” in this article to ensure rigorous and scientific language. Thank you again for your valuable feedback.

Reviewer 2 Report

Comments and Suggestions for Authors

Authors make proper corrections according to reviewer remarks and

I suggest to publish the paper as it is.

Author Response

Comment 1:

Authors make proper corrections according to reviewer remarks and I suggest to publish the paper as it is.

Response 1:

We appreciate the positive comments made by the reviewer and sincerely value your suggestions, which have improved our manuscript and helped it meet the standards for publication.

Reviewer 3 Report

Comments and Suggestions for Authors

Main concern on the first review was the handling of SF6. Though the authors addressed this in Line 113: "Furthermore, the outlet of the reactor is seamlessly integrated with a collection device equipped with activated carbon, ensuring comprehensive absorption of the introduced SF6."

The authors' response regarding the use of activated carbon does not adequately address the primary concern related to the handling of SF6. Activated carbon is not effective at absorbing SF6, as SF6 is an inert gas with very low reactivity and a relatively large molecular size, making it difficult for activated carbon to adsorb it efficiently. Activated carbon is typically used to adsorb organic compounds, odors, and certain other gases, but it is not suitable for SF6, which does not interact significantly with the carbon's surface.

To safely handle SF6, the authors should consider alternative methods:

1.     Gas Recapture: Implementing a gas recovery system designed to capture SF6 from the exhaust stream, allowing it to be recycled for further use.

2.     Neutralization: Utilizing technologies that can break down SF6 into less harmful components, though this typically requires high-energy processes such as plasma or laser methods.

3.     Safe Disposal: Ensuring SF6 is stored and disposed of by professional hazardous waste services that comply with environmental regulations.

Comments on the Quality of English Language

No comments for the 2nd review

Author Response

Comments 1:

Main concern on the first review was the handling of SF6. Though the authors addressed this in Line 113: "Furthermore, the outlet of the reactor is seamlessly integrated with a collection device equipped with activated carbon, ensuring comprehensive absorption of the introduced SF6."

The authors' response regarding the use of activated carbon does not adequately address the primary concern related to the handling of SF6. Activated carbon is not effective at absorbing SF6, as SF6 is an inert gas with very low reactivity and a relatively large molecular size, making it difficult for activated carbon to adsorb it efficiently. Activated carbon is typically used to adsorb organic compounds, odors, and certain other gases, but it is not suitable for SF6, which does not interact significantly with the carbon's surface.

To safely handle SF6, the authors should consider alternative methods:

  1. Gas Recapture: Implementing a gas recovery system designed to capture SF6 from the exhaust stream, allowing it to be recycled for further use.
  2. Neutralization: Utilizing technologies that can break down SF6 into less harmful components, though this typically requires high-energy processes such as plasma or laser methods.
  3. Safe Disposal: Ensuring SF6 is stored and disposed of by professional hazardous waste services that comply with environmental regulations.

Response 1:

We appreciate the valuable comments provided by the reviewer. The three solutions you provided have broadened our horizons and given us a deeper understanding of the recycling and utilization of SF6. Actually, the collection method we adopted in the experiment is similar to the third strategy you mentioned above. We first collect SF6 using a closed container to ensure that it is not directly released into the air, and then hand it over to a professional hazardous waste agency that complies with environmental regulations for proper disposal. Furthermore, your insightful comments made us realize that activated carbon is not a suitable material for adsorbing SF6. Therefore, we have deleted this statement from the original text and revised in lines 113-116, as follows:

Notably, the outlet of the reactor is equipped with a seamlessly integrated collection device, ensuring effective recovery of the introduced SF6 gas. The recovered SF6 is transferred to a professional hazardous waste agency to ensure that it is not directly released into the air.

In addition, we kindly ask for your understanding regarding any inaccuracies in our expression concerning the recovery of SF6. As the topic of this paper is on the study of the impact of plasma treatment on surface flashover of epoxy resin, we acknowledge that there are indeed some deficiencies in our research regarding the recycling and utilization of SF6. Inspired by your Knowledgeable comment, we have carefully reviewed the literature and summarized the current main methods for the recycling and utilization of SF6.

(i) Using plasma, electrical discharge, spark, or microwave radiation to decompose SF6, typically achieves nearly complete decomposition. However, such treatment is prone to generating a range of corrosive and toxic gases, including SOF4, SO2F2, SF4, SOF2, and so on [1-3].

(ii) Using inorganic materials for adsorption, such as silicalite, zeolite, carbon nanotubes, and pillared clay. However, these materials typically present a very limited adsorption capacity for SF6, resulting in low removal efficiency [4-6]. (Similarly, we have gained a deeper understanding that activated carbon is also not suitable for effectively adsorbing SF6)

(iii) Using catalytic decomposition, including photodegradation and thermal degradation, some catalysts are able to decompose nearly 100% of SF6, producing fewer corrosive and toxic gases than the (i) method. However, the process steps are relatively cumbersome [7-9].

Regarding above research methods and the options you provided, we have comprehensively considered that direct collection and delivery to professional hazardous waste agency is the simplest disposal approach for this work. In the future, we will consider researching more convenient and environmentally friendly treatment measures, and we look forward to receiving more guidance from you.

  1. Tsai, C.-H.; Shao, J.-M. Formation of fluorine for abating sulfur hexafluoride in an atmospheric-pressure plasma environment. Journal of Hazardous Materials 2008, 157, 201-206, doi:https://doi.org/10.1016/j.jhazmat.2008.01.010.
  2. Radoiu, M.; Hussain, S. Microwave plasma removal of sulphur hexafluoride. Journal of Hazardous Materials 2009, 164, 39-45, doi:https://doi.org/10.1016/j.jhazmat.2008.07.112.
  3. Kurte, R.; Heise, H.M.; Klockow, D. Analysis of spark decomposition products of SF6 using multivariate mid-infrared spectrum evaluation. Journal of Molecular Structure 1999, 480-481, 211-217, doi:https://doi.org/10.1016/S0022-2860(98)00642-5.
  4. Chiang, Y.-C.; Wu, P.-Y. Adsorption equilibrium of sulfur hexafluoride on multi-walled carbon nanotubes. Journal of Hazardous Materials 2010, 178, 729-738, doi:https://doi.org/10.1016/j.jhazmat.2010.02.003.
  5. Furmaniak, S.; Terzyk, A.P.; Gauden, P.A.; Kowalczyk, P. Simulation of SF6 adsorption on the bundles of single walled carbon nanotubes. Microporous and Mesoporous Materials 2012, 154, 51-55, doi:https://doi.org/10.1016/j.micromeso.2011.09.030.
  6. Chuah, C.Y.; Lee, Y.; Bae, T.-H. Potential of adsorbents and membranes for SF6 capture and recovery: A review. Chemical Engineering Journal 2021, 404, 126577, doi:https://doi.org/10.1016/j.cej.2020.126577.
  7. Song, X.; Liu, X.; Ye, Z.; He, J.; Zhang, R.; Hou, H. Photodegradation of SF6 on polyisoprene surface: Implication on elimination of toxic byproducts. Journal of Hazardous Materials 2009, 168, 493-500, doi:https://doi.org/10.1016/j.jhazmat.2009.02.047.
  8. Kashiwagi, D.; Takai, A.; Takubo, T.; Yamada, H.; Inoue, T.; Nagaoka, K.; Takita, Y. Catalytic activity of rare earth phosphates for SF6 decomposition and promotion effects of rare earths added into AlPO4. Journal of Colloid and Interface Science 2009, 332, 136-144, doi:https://doi.org/10.1016/j.jcis.2008.12.003.
  9. Wu, Y.; Gao, P.; Li, Y.; Yang, Z.; Wan, K.; Zhang, X. Degradation of SF6 by dielectric barrier discharge cooperating with TiO2 photocatalysis: Insights into the reaction mechanism. Applied Surface Science 2024, 660, 159957, doi:https://doi.org/10.1016/j.apsusc.2024.159957.
